# Chinese Consumers’ Attitudes toward and Intentions to Continue Using Skill-Sharing Service Platforms

**DOI:** 10.3390/bs14090765

**Published:** 2024-09-02

**Authors:** Yaxiao Chen, Mi Hyun Ryu

**Affiliations:** 1Department of International Business and Commerce, Graduate School, Konkuk University, 120 Neungdong-ro, Gwangjin-gu, Seoul 05029, Republic of Korea; chenyaxiao0813@gmail.com; 2Department of Global Business, Konkuk University, 120 Neungdong-ro, Gwangjin-gu, Seoul 05029, Republic of Korea

**Keywords:** sharing economy, skill-sharing service platform, UTAUT model, attitude, intention to continue using

## Abstract

The sharing economy has rapidly transformed traditional consumption patterns worldwide. The emergence of skill-sharing services—which allow individuals to share their skills, abilities, and time through online platforms—has recently garnered attention. In China, the demand for skill-sharing services continues to grow, as these services effectively meet consumer needs. To understand this growing demand, this study aims to explore users’ attitudes and intentions toward the use of skill-sharing service platforms in the Chinese market. A survey was conducted that incorporated 500 Chinese users who had used skill-sharing service platforms over the previous three months. A total of 409 datasets were analyzed, using structural equation modeling and multiple group analysis, in AMOS 24.0. The results showed that performance expectancy, effort expectancy, social influence, facilitating conditions, and self-efficacy positively influenced users’ attitudes toward skill-sharing services, while privacy, functionality, and safety risks negatively affected these attitudes. Users’ attitudes toward skill-sharing services significantly enhanced their intentions to continue using them, with the level of trust playing a crucial moderating role between attitude and the intention to continue using these services. These findings provide a significant theoretical and practical foundation for the further development of skill-sharing service platforms, the optimization of marketing strategies, and future research.

## 1. Introduction

Sharing is the oldest form of consumption [1]. Although the concept of sharing is well-established, the development of information and communication technology has transformed traditional offline sharing into online sharing, expanding the range of participants globally [2]. As an emerging economic model, the sharing economy has significantly changed the existing socio-economic system. To date, the sharing economy has encompassed almost all types of online and offline person-to-person (P2P) economic activities and has become a crucial part of today’s business patterns and consumer lifestyle trends [3]. Its scope has increasingly expanded to include goods, healthcare services, space-sharing, and pet care. The surge in sharing platforms has allowed people to access resources that were previously unavailable to them, drastically changing consumption habits and lifestyles [4].

Notably, online work platforms have found growing acceptance due to their greater flexibility, lower entry barriers, and potential to supplement basic income [5]. These platforms connect consumers or service requesters with service providers who can perform the requested tasks, promoting diversification and inclusivity in the labor market while enabling consumers to have their service needs met rapidly [6]. These skill-sharing services, as a form of the sharing economy, involve the sharing of intangible assets such as knowledge, skills, and time. They are defined as services that allow others to use an individual’s personal time, skills, and knowledge through a platform [6]. In this study, a skill-sharing service platform is defined as a service that shares idle skills, abilities, or time resources with individuals who need them via a platform.

Skill-sharing service platforms face vast market demand for meeting a wide variety of consumer needs, improving consumption experiences, providing economical choices to consumers, and expanding career development opportunities [7]. User perceptions that skill-sharing service platforms significantly enhance convenience in daily life and are highly useful encourage other consumers to use them [8].

In China, skill-sharing services based on human assets such as skills, abilities, or time have rapidly developed, significantly impacting overall economic and social development [9]. Services provided through Chinese sharing service platforms mainly include household and errand services, such as help with moving, household management, prenatal and childcare, appliance repair, shopping and household labor, and queuing. Representative skill-sharing service platforms in China include JD jiazheng, Suning bangke, 58 tongcheng, and UU paotui.

According to a report released by iiMedia [10], the Chinese household service market grew nearly threefold from a market size of CNY 277.6 billion in 2015 to CNY 1 trillion 89 billion in 2022. In addition, the market size of China’s errand economy—which stood at CNY 13.1 billion in 2021—is projected to reach CNY 66.4 billion by 2025 [11]. Despite the growing use and importance of skill-sharing services among Chinese consumers, there is a scarcity of relevant research. Therefore, this study analyzes related consumer attitudes and intentions to continue to use these services.

This study employs the unified theory of acceptance and use of technology (UTAUT) model, which has demonstrated an explanatory power of 70% for variance in consumer intentions in terms of technology acceptance [12]. By incorporating variables such as perceived risk and self-efficacy, which are frequently used in UTAUT extensions [13], as well as an altruistic tendency, which plays a significant role in collectivist societies [14], this study seeks to explore how these variables affect attitudes toward and intentions to continue using skill-sharing service platforms.

Moreover, sharing economy platforms integrate online networks with real-world transactions, emphasizing the importance of trust in interactions among consumers, service providers, and platforms [15]. Therefore, this study aims to examine whether users’ levels of trust in skill-sharing service platforms moderate their attitudes and intentions to continue using these services. To address these objectives, this study attempts to answer the following research questions regarding skill-sharing service platforms:

RQ1. Which factors influence consumer attitudes toward skill-sharing service platforms, and how do these attitudes further impact their intentions to continue using them?

RQ2. Does trust in skill-sharing service platforms have a moderating effect?

By discussing these key questions, this study provides theoretical support for the marketing strategies of skill-sharing service platforms and also offers concrete evidence for these strategies through empirical research. Concurrently, the results of this study serve as foundational data to enable consumers to use platforms efficiently.

## 2. Theoretical Background and Hypothesis Development

### 2.1. Unified Theory of Acceptance and Use of Technology (UTAUT) Model

The UTAUT model was proposed by Venkatesh et al. [16] by integrating theories from various fields, including the theory of planned behavior, the motivational model, the innovation diffusion theory, the social cognitive theory, and combined TAM and TPB, which have been developed and used to explain the use intention and use of new technologies. Despite its recent introduction, the UTAUT model has garnered significant worldwide attention from scholars due to its explanatory power of up to 70% in the research field of user intention [12]. It is widely used as a core theory to explain issues related to technology acceptance in various fields such as e-commerce, economic management, and information systems [17]. Tamilmani et al. [13] reviewed studies on UTAUT and suggested adding individual variables, such as attitude and trust. Accordingly, this study adds attitude as a predictor of the consumers’ intentions to continue using skill-sharing platforms. The UTAUT model is the core framework, with the added variables of attitudes and intention to continue using the platforms.

The UTAUT model explains consumer behaviors using four variables: performance expectancy, effort expectancy, social influence, and facilitating conditions. Performance expectancy is defined as the degree to which users believe that applying a particular system or technology will enhance their performance in regards to specific tasks [16]. Similar to perceived usefulness, users perceive that they can initiate sharing actions anytime and anywhere in order to meet their needs. In this process, it is crucial that users perceive its usefulness and experience strong functional benefits when using the technology [18]. In this study, performance expectancy is defined as the advantages, benefits, and time-saving aspects users perceive when using skill-sharing service platforms. The perception that skill-sharing services enhance the efficiency of daily life and are beneficial to the user will lead to more favorable attitudes toward using the service platform.

Effort expectancy refers to the ease or difficulty users experience when using new technology, i.e., the level of effort users are willing to expend to use the technology or service [16]. Users tend to reject a technology or service if they perceive the process as complex or costly to learn [18]. In this study, effort expectancy is defined as the level of ease consumers experience when using a skill-sharing service. The less effort users need to expend to understand and learn new technology, the more likely they are to adopt it.

Social influence is defined as the degree to which consumers perceive that their significant others believe they should use a new technology or service [16]. Users, as members of society, are easily influenced by their social environment (friends, family, and peers) and by promotions and recommendations from various media sources [19]. Group encouragement of the sharing economy leads to greater individual acceptance [18]. In the context of this study, social influence is defined as the extent to which consumers are swayed by others’ opinions and suggestions regarding the use of skill-sharing service platforms, as well as the influence of media such as newspapers and broadcasts.

The facilitating conditions variable is defined as the degree to which consumers believe that there is organizational and technical infrastructure to support the use of a new information technology or system [16]. Research has demonstrated that when consumers believe that there are sufficient facilitating conditions for using a new technology or service, their burden of adoption is alleviated, and facilitating conditions encourage the use of information technology [20]. In this study, facilitating conditions are defined as the extent to which consumers believe they can receive environmental and technical support when using a skill-sharing service platform, including access to relevant resources and information.

Driven by technological advancement and market expansion, UTAUT has been increasingly utilized in various fields of research. It is also widely used in the sharing service domain to predict consumer attitudes. In a study on shared autonomous vehicles, Yuen et al. [8] found that performance expectancy and effort expectancy have a significant positive impact on attitudes. In a study on car-sharing, Cylwik et al. [21] also confirmed the positive effect of performance expectancy on user attitudes. Elnadi and Gheith [22] and Goel and Haldar [23] also conducted studies on car-sharing and found positive associations between perceived usefulness, usability, social influence, and consumer attitudes. In their empirical research on the Airbnb platform, Tamilmani et al. [24] revealed that effort expectancy, social influence, and facilitating conditions shape consumer attitudes toward accommodation-sharing technology.

Based on these previous studies, we propose the following hypotheses:

**H1:** *Performance expectancy will positively affect attitudes toward skill-sharing service platforms*.

**H2:** *Effort expectancy will positively affect attitudes toward skill-sharing service platforms*.

**H3:** *Social influence will positively affect attitudes toward skill-sharing service platforms*.

**H4:** *Facilitating conditions will positively affect attitudes toward skill-sharing service platforms*.

### 2.2. Perceived Risk

Despite its widespread application across various fields, UTAUT theory still presents several limitations [25]. This has prompted researchers to expand and refine the model by integrating it with other theories or incorporating additional variables [20]. Tamilmani et al. [26] conducted a meta-analysis of UTAUT extensions and found that the most common form of expansion was the addition of constructs, including perceived risk. Consequently, this study incorporates perceived risk as a variable to assess the negative aspects of skill-sharing services within the UTAUT model.

Perceived risk is a construct introduced by Bauer in 1960 [27]. It has been widely adopted in marketing and psychology research to investigate consumer purchasing behaviors [28]. If the outcome of the use of a selected product is uncertain, users feel uncomfortable. Their purchasing behavior inherently carries risk due to this uncertainty [29]. Perceived risk can be defined as the uncertainty consumers experience when purchasing a product or service, as well as their concerns about potential negative consequences [30]. Furthermore, consumer attitudes toward e-commerce platforms can be significantly influenced by their perceived risk [31]. Therefore, when predicting consumer attitudes and behavioral intentions, perceived risk should be considered as a crucial variable. In this study, perceived risk is defined as the degree to which consumers perceive uncertainty or potential losses resulting from their choices when using skill-sharing service platforms.

Furthermore, perceived risk is multidimensional and can be classified into several categories. Yang et al. [32] divided the risks associated with online transactions into two main categories: system-related and transaction-related risks. System-related risks encompass functional, safety, time, and social risks, while transaction-related risks include financial, privacy, service, and psychological risks.

Among these, financial, privacy, and functional risks are the most relevant to the uncertainties of online transactions and are widely utilized in research related to online services [33]. Unlike traditional online services, skill-sharing platforms operate as an O2O (online-to-offline) sharing economy business model, providing both online and offline services. This hybrid approach can introduce a series of safety-related risks that may lead to serious social issues [34]. Safety risks have become a prevalent concern in the sharing service economy environment [35]. As skill-sharing service platforms fall under the O2O category, users may encounter online and offline risks. These risks include privacy risks related to the leakage of personal information [33], financial risks that consumers perceive as potential economic losses or unfavorable outcomes [36], functional risks associated with the purchased service not meeting consumer expectations [37], and physical and property safety risks that users may face during the service process [22]. Therefore, this study will examine four types of risks most relevant to skill-sharing service platforms: privacy, financial, functional, and safety risks.

Kim and Lee [36] examined perceived risk in the sports-sharing economy sector and found that functional, financial, and safety risks negatively affect consumer attitudes. Similarly, Hasan et al. [37], in their study on car-sharing, found that privacy, safety, and functional risks imposed significant negative effects on attitudes. Elnadi and Gheith [22] corroborated these findings, demonstrating that safety and privacy risks negatively influence attitudes toward car-sharing. These studies collectively emphasize that users’ perceived risks related to privacy, financial, functional, and safety concerns are crucial factors influencing decision making and attitudes in the sharing economy. 

Based on these previous findings, we propose the following hypotheses:

**H5:** *Perceived privacy risks associated with skill-sharing service platforms will negatively affect attitudes*.

**H6:** *Perceived financial risks associated with skill-sharing service platforms will negatively affect attitudes*.

**H7:** *Perceived functional risks associated with skill-sharing service platforms will negatively affect attitudes*.

**H8:** *Perceived safety risks associated with skill-sharing service platforms will negatively affect attitudes*.

### 2.3. Self-Efficacy

An individual’s self-efficacy beliefs play a crucial role in addressing consumer issues and have gained prominence in marketing research [38]. As a fundamental determinant of behavior in social psychology, self-efficacy directly and indirectly influences individuals’ actions [39]. Therefore, this investigation of the factors that shape consumer attitudes toward skill-sharing service platforms incorporates self-efficacy. 

Bandura [40], who first introduced the concept of self-efficacy, described it as an individual’s belief in their ability to organize and execute the actions required to achieve desired accomplishments. It acts as a form of self-assessment and plays a very important role in determining intentional behavior [41]. In this study, self-efficacy is defined as the degree of belief consumers have in their ability to effectively use skill-sharing service platforms to achieve their goals.

Chen et al. [41], in their study on knowledge-sharing services, found that self-efficacy significantly influences consumer attitudes. Similarly, in their correlational study on the sharing economy, Zhu et al. [42] demonstrated that self-efficacy positively affects both attitudes and behavioral intentions.

Based on these prior findings, we propose the following hypothesis:

**H9:** *Self-efficacy will have a positive influence on user attitudes toward skill-sharing service platforms*.

### 2.4. Altruistic Tendency

From the perspective of social exchange theory, participation in collaborative consumption can be considered prosocial behavior due to its positive impact on others [43]. Altruistic tendency refers to actions undertaken for the satisfaction or well-being of others, without expecting any reward in return. It is a voluntary behavior or psychological trait in which individuals incur costs to provide economic benefits to others [44]. This altruistic tendency is an integral part of the psychological makeup of sharing economy service providers and enhances overall societal efficiency, as consumers achieve their desired outcomes while simultaneously providing economic benefits to others [14]. When consumers use sharing services, they experience a unique psychological state in which their consumption behavior is not indulged in solely for personal enjoyment, but is also a social action that creates employment opportunities for others [45]. Within the scope of this study, altruistic tendency is defined as an individual’s selfless inclination to prioritize helping others in difficult situations, without considering personal gain or loss, motivated by a sense of meaning derived from helping others.

In a study on the sharing economy, Bucher et al. [46] argued that moral motivation for sharing significantly influences attitudes. This moral motivation encompasses elements such as sustainability and altruism. Similarly, Ratilla et al. [47], in a study on the sharing economy platform, observed that as the sharing economy implements a type of pro-social behavior, consumers’ altruistic tendency influences their attitudes.

Based on these prior findings, we propose the following hypothesis:

**H10:** *Altruistic tendency will positively influence attitudes toward skill-sharing service platforms*.

### 2.5. Attitude and Intentions to Continue Use 

An attitude is a learned tendency to respond consistently, either favorably or unfavorably (positively or negatively) toward a given situation [48]. Consequently, attitudes influence users’ behavioral intentions, which in turn affect their actual behaviors. Measuring attitudes, along with intentions and satisfaction, provides insight into predicting consumer behavior [49]. In this study, attitude is defined as the favorable or unfavorable response of consumers toward using skill-sharing service platforms.

The intention to continue use is the extent to which a user plans to keep using a particular product or service. When a product or service meets or exceeds expectations, users are motivated to continue using it due to the positive emotional experiences it provides, such as enjoyment [50]. In this study, intention to continue use is defined as the users’ intention to consistently and more frequently utilize skill-sharing service platforms in the future.

Attitude is a key determinant of the intention to continue use, with consumers’ attitudes positively influencing their behaviors [51]. The inclusion of attitude significantly enhances the explanatory power of theoretical models. For instance, in a study examining UTAUT, the explanatory power for behavioral intention increased from 38% to 45% when attitude was considered [52]. Hwang and Griffiths [43], in their study on millennials’ use of collaborative consumption services, found that young consumers’ positive attitudes toward these services increased their intentions to purchase. Similarly, Seyed and Reynolds [53] demonstrated that attitude positively affects intention to continue use in the sharing economy. Ratilla et al. [47] corroborated these findings in their study on sharing economy platforms, confirming that the level of consumer attitude positively influences sharing intention.

Based on these prior findings, we propose the following hypothesis:

**H11:** *A positive attitude toward skill-sharing service platforms will positively influence the intention to continue using these services*.

### 2.6. Trust

Trust is a crucial factor that shapes consumer behavior and is essential for socio-economic interactions. It is a driver of consumer participation [36]. In the context of sharing economy services, trust plays a vital role in influencing consumer attitudes and refers to the belief that consumers can rely on a service consistently over time [54]. Trust is one of the primary factors facilitating transactions between consumers and skill providers, as well as between consumers and platforms. Consequently, this study incorporates trust as a critical variable when examining the sharing economy.

Trust has been conceptualized in two primary dimensions, as elucidated in the studies by Ganesan and Hess [55] and Oliveira et al. [56]. The first dimension, credibility, is predicated based on the core partner’s intention and ability to fulfill commitments. It encompasses trust in the partner’s specific work capabilities, the reliability of delivered products and services, and the predictability of business-related actions. The second dimension, benevolence, is based on the core partner’s qualities and intentions. It transcends purely self-interested motives, demonstrating genuine concern and care for the partner. In the context of the sharing economy, trust can be defined as the confidence that transactions will be completed successfully without misunderstanding, harm, or exploitation [57]. High levels of consumer trust in sellers facilitate smoother purchasing processes. Given the various potential risks associated with the sharing economy, users’ trust in sharing economy platforms determines their willingness to participate [58]. For this study, trust is defined as the degree of confidence in the platform itself, as well as in the accuracy and reliability encountered when using skill-sharing service platforms.

Consumer trust has become a key factor in the success of services, effectively reducing uncertainty in decision-making processes and enhancing a company’s market competitiveness [59]. The importance of trust is particularly pronounced in regards to online services [60]. As consumers cannot directly interact with products or meet service providers face-to-face, they must rely on brand reputation, user reviews, positive online experiences, information quality, and safety to make decisions [61]. Trust, in this context, not only provides consumers with a sense of security in online transactions but also directly influences whether or not they decide to engage in transactions in uncertain environments [62]. Therefore, building online trust benefits both consumers and companies through close interactions, transactions, and exchanges [63]. For online service providers, trust has become a key factor in retaining existing customers [64].

Recent research indicates that consumer trust in sharing economy platforms positively influences usage intentions [65]. Ye et al. [66], in their study on accommodation-sharing, corroborated that trust has a moderating effect between satisfaction and intention to continue use. Trust was also identified as a significant moderator between attitudes toward new technologies and their intention to adopt [67]. Consequently, trust in skill-sharing service platforms can facilitate the relationship between attitude and intention. Consumers with high trust in the platform are more likely to believe that continued use of skill-sharing service platforms will yield positive outcomes and have higher expectations for such results. Increased trust in the platform will encourage consumers to continue using skill-sharing service platforms due to their anticipation of positive and feasible outcomes.

Based on this, the following hypothesis is proposed:

**H12:** *Trust in skill-sharing service platforms will moderate the relationship between attitude and the intention to continue use*.

### 2.7. Model Construction

As illustrated in Figure 1, this study examines how attitude is influenced by performance expectancy, effort expectancy, social influence, facilitating conditions, perceived risk, self-efficacy, and altruistic tendency. Additionally, it investigates the impact of attitudes on the intention to continue use and evaluates the moderating effect of trust.

## 3. Research Methodology

### 3.1. Survey Instrument

To test the proposed hypotheses, an online survey was conducted. Although probability sampling is generally considered the standard method for obtaining a representative sample, allowing for effective statistical inference across all variables, issues such as frame coverage deficiencies and non-response make it increasingly difficult to collect a strict probability sample. Additionally, the cost of implementing rigorous probability sampling has been rising [68]. Thus, this study did not use traditional probability sampling, instead opting for voluntary sampling as an alternative. Voluntary sampling is a non-probability sampling method in which survey samples are selected from among voluntary and qualified potential respondents within the target population. Before implementing the sample design, the research team announced the intent of the survey and provided potential respondents with sufficient time to decide whether or not to participate [69]. However, voluntary sampling also presents significant limitations, primarily due to selection bias, as it may not fully represent the entire target population. Despite these limitations, voluntary sampling remains a viable and appropriate choice for researchers when obtaining a random sample is challenging [70]. Therefore, this study employed voluntary sampling. 

Data were collected through an online survey platform in China, with the questionnaire comprising 50 items. Four demographic variables were included: gender, age, education level, and monthly income. The key variables of the research model were measured using a 5-point Likert scale (1 = strongly disagree, 5 = strongly agree). Table 1 presents the measurement items and their sources. To ensure the accurate translation of the measurement items, a professional translator reviewed the questionnaire.

### 3.2. Data Collection

To ensure the questionnaire’s readability and comprehensibility, a preliminary survey was conducted with 63 Chinese respondents before the main survey was launched. The feedback indicated that some participants did not fully grasp the concept of a “skill-sharing service platform”. Consequently, the researcher added a detailed explanation at the beginning of the questionnaire to enhance participants’ understanding of the concept.

The main survey targeted Chinese users aged 20 and above who had utilized skill-sharing service platforms within the previous three months. The survey was conducted from 1–15 April 2024, using a popular online survey platform in China. A total of 500 questionnaires were distributed, and after eliminating invalid responses, 409 valid responses were obtained, yielding a response rate of 81.8%. The demographic characteristics of the respondents are presented in Table 2.

### 3.3. Data Analysis

Following data collection, SPSS 26.0 and AMOS 24.0 software were employed for data analysis. To verify the validity and reliability of the latent variables, a pooled confirmatory factor analysis (CFA) was conducted on the final measurement model, which comprised 46 items. Subsequently, a two-step testing process was applied to the structural model. The first step involved verifying the relationships between variables, while the second step consisted of a multi-group comparison analysis to examine the moderating role of trust.

## 4. Results

### 4.1. Confirmatory Factor Analysis

Table 3 presents the factor loadings, Cronbach’s alpha, composite reliability (CR), and average variance extracted (AVE) for the variables. For reliability testing, a Cronbach’s alpha value of 0.7 or above was considered acceptable [72]. In this study, Cronbach’s alpha values ranged from 0.797–0.929, demonstrating sufficient reliability.

To assess convergent validity, the standardized factor loadings of all items were examined and found to exceed the minimum criterion of 0.5. The composite reliability coefficients ranged from 0.799–0.929, surpassing the threshold of 0.7, while the AVE ranged from 0.547–0.749, exceeding the recommended value of 0.5. These results confirm that this study’s measurement model demonstrates adequate convergent validity [73].

Discriminant validity was assessed following the guidelines proposed by Fornell and Larcker [74]. Specifically, the square root of the AVE for each latent variable should exceed its correlations with other latent variables. As shown in Table 4, the square root of the AVE is greater than the correlation coefficients with other factors, indicating acceptable discriminant validity and the absence of multicollinearity issues in the measurement model used in this study.

Moreover, the model fit indices of the measurement model were satisfactory (Chi-square = 1166.861, df = 911, χ^2^/df = 1.281; *p* = 0.000, GFI = 0.894; AGFI = 0.875; CFI = 0.975; RMSEA = 0.026), with all fit indices falling within the recommended thresholds. A model is considered acceptable when χ^2^/df < 3, GFI > 0.8, AGFI > 0.8, CFI > 0.9, and RMSEA < 0.08 [75,76].

### 4.2. Hypothesis Testing

The model fit indices (Chi-square = 1032.405, df = 804, χ^2^/df = 1.284; *p* = 0.000, GFI = 0.899; AGFI = 0.881; CFI = 0.976; RMSEA = 0.026) met the recommended criteria, indicating that the overall model fit is acceptable. Table 5 outlines the results of the hypothesis testing.

Hypotheses H1 through H4 were tested as the first set of hypotheses. As expected, performance expectancy (β = 0.173, *p* < 0.001), effort expectancy (β = 0.160, *p* < 0.001), social influence (β = 0.161, *p* < 0.01), and facilitating conditions (β = 0.149, *p* < 0.01) all positively influenced attitude, thus supporting H1 through H4. These findings align with the hypotheses of the UTAUT model, demonstrating that performance expectancy, effort expectancy, social influence, and facilitating conditions all have a positive impact on users’ attitudes toward skill-sharing service platforms.

Hypotheses H5 through H8 were tested to examine the effects of various risks on attitudes toward using skill-sharing service platforms. Financial risk (β = 0.050, *p* > 0.05, n.s.) did not significantly affect attitudes toward using skill-sharing service platforms, thus rejecting H6. However, privacy risk (β = −0.114, *p* < 0.05), functional risk (β = −0.137, *p* < 0.01), and safety risk (β = −0.136, *p* < 0.001) all had significant negative effects, supporting H5, H7, and H8, respectively. These results indicate that among the various risks examined, privacy, functional, and safety risks are the primary factors limiting positive attitudes toward skill-sharing service platforms. Users may maintain psychological barriers due to potential negative situations that could arise while using these platforms, thereby reducing their positive attitudes toward them.

Furthermore, self-efficacy demonstrated a strong positive impact on attitude (β = 0.204, *p* < 0.001), supporting H9. Contrary to expectations, altruistic tendency did not significantly influence attitude (β = −0.038, *p* > 0.05, n.s.), leading to the rejection of H10. Additionally, attitude exhibited a positive influence on intention to continue use (β = 0.654, *p* < 0.001), supporting H11.

### 4.3. Moderation Analysis

The moderating effect of trust was examined using multi-group SEM analysis. Following the method proposed by Cheng et al. [77], the entire sample was divided into two groups—a low-trust and a high-trust group—based on the mean trust score (3.665). An χ^2^ difference test was then conducted between the constrained and unconstrained models, yielding significant results (Δχ^2^ = 7.910, Δdf = 1), thus supporting H12, as shown in Table 6. Although attitude positively influenced intention to continue use in both groups, the effect coefficient was higher in the high-trust compared to the low-trust group. Users in the high-trust group tended to believe that the platform can consistently deliver high-quality services due to their high level of trust in the platform, which in turn strengthens their attitude toward continuing to use the platform. Trust increases expectations of the platform’s future performance and enhances the impact of attitude on the intention to continue using the service. When users exhibit higher trust, they are more likely to continue using the service. In contrast, in the low-trust group, even if users have a positive attitude, the lack of trust in the platform may result in a relatively lower likelihood of this attitude translating into an intention to continue use. This finding suggests that for users with higher trust levels, the positive influence of attitude on intention to continue use is more pronounced. It also highlights the important role of trust as a moderating variable in strengthening the effect of attitude on behavioral intention and better reflects how crucial trust is in the user decision-making process.

## 5. Conclusions and Discussion

### 5.1. Theoretical Implications 

This study distinguishes itself from previous research in four key aspects. First, while prior studies have predominantly applied the UTAUT theory to examine intention to use, researchers have consistently recommended incorporating individual characteristic variables (such as attitude and trust) into the UTAUT model [13]. In response, this study enhances and broadens the theoretical framework by including attitude as a crucial predictor of continued intention to use. As hypothesized, performance expectancy, effort expectancy, social influence, and facilitating conditions significantly affected attitude, which in turn positively influenced continued use intention. These findings reaffirm the UTAUT theory’s applicability and validity in sharing economy research, offer new insights into skill-sharing service platforms, and provide novel research avenues and frameworks for attitude studies.

Second, previous research on the sharing economy has primarily focused on its positive aspects, with limited attention given to negative elements such as resistance and potential risks [78]. This study addresses this gap by categorizing perceived risks into four types: privacy, financial, functional, and safety risks, and examines their respective effects on user attitudes. The results demonstrate that higher perceived risks associated with skill-sharing service platforms lead to less favorable attitudes, underscoring the need to mitigate these risks to foster positive perceptions. These findings highlight the importance of considering potential negative impacts, such as risks, when evaluating attitudes toward these platforms. By incorporating both positive influencing factors and negative elements, like perceived risks, this study contributes to a more comprehensive approach to attitude prediction. Additionally, this research provides valuable insights by identifying and analyzing the specific components of perceived risks associated with skill-sharing service platforms. 

Third, to enhance our understanding of attitude formation in the sharing economy, this study incorporates two key psychological characteristics: self-efficacy and altruistic tendency [39,45]. The findings from our research on skill-sharing service platforms illuminate the specific role of consumer psychology in technology-driven sharing economies. This contribution enriches the existing literature on the sharing economy and offers fresh insights into the mechanisms shaping attitudes within this context. Furthermore, it broadens the scope of factors known to influence consumer attitudes in this domain.

Finally, this study introduces trust as an additional individual characteristic variable in order to analyze its moderating role. The research demonstrates that trust positively moderates the relationship between attitude and intention to continue use. When users trust the platform and believe it can deliver high-quality services, their positive attitude is more likely to result in the intention to continue using it. Consequently, this study offers a fresh perspective on the impact of trust but also expands the role of trust within the UTAUT model, highlighting its complex influence on user decision-making processes.

### 5.2. Practical Implications

Recent years have witnessed the emergence of skill-sharing services as a new form of service within the sharing economy. These services inherit the core characteristics of the sharing economy, transforming people’s lifestyles [6] and promoting the effective use of resources, maximizing market efficiency. This has greatly benefited consumers and brought convenience to their lives [79]. However, previous research on the sharing economy has primarily focused on transportation, accommodation, and tourism, with relatively few studies dedicated to investigating skill-sharing services. As a result, companies lack reference points for effective market strategy formulation. Therefore, this study provides implications for the management of skill-sharing service platforms and offers insights to platform operators and market strategists, helping them gain a deeper understanding of user needs and consumer motivations. Additionally, it will support the further development of the sharing economy.

The findings of this study reveal five focus areas associated with skill-sharing service platforms. First, performance expectancy, effort expectancy, social influence, and facilitating conditions are crucial factors in shaping consumer attitudes toward skill-sharing service platforms. Consistent with Tamilmani et al.’s study [24] of the accommodation-sharing industry, performance expectancy positively influences attitudes. It is essential to make consumers aware of the benefits the platform can provide [80]. Therefore, skill-sharing service platforms should widely communicate the benefits of using these platforms, particularly in regards to saving time. In line with Yuen et al.’s study [8] in the car-sharing sector, effort expectancy was found to positively influence attitudes. Given the user preference for easy-to-use platforms [81], skill-sharing service platforms should optimize user interfaces, making them intuitively understandable and easy to use by simplifying the usage process, effectively reducing users’ learning costs and usage barriers. Additionally, consistent with the findings of Goel and Haldar [23] regarding the car-sharing sector, social influence positively impacted attitudes. Given the vital roles of social recognition and word-of-mouth promotion for sharing service platforms [24], skill-sharing service platforms should actively showcase the effectiveness of their services across various media channels. They should also encourage existing users to share their experiences with others or on social media, thereby enhancing the platform’s appeal. Furthermore, as demonstrated by Ye et al. [82] in the sharing economy, facilitating conditions significantly influence attitudes. Users need to understand how to use the service effectively, and the platform should provide timely assistance when users face difficulties or technical issues [82]. Therefore, if skill-sharing service platforms offer efficient technical support and customer service through various channels, as well as provide relevant usage information, this can significantly improve the consumer experience. These factors are crucial to enhance consumer attitudes toward skill-sharing service platforms.

Second, as perceived risk can significantly influence people’s attitudes, decisions, and behaviors [15], we examined the impact of perceived risk on attitudes toward skill-sharing service platforms. The findings revealed that financial risk does not significantly affect usage attitudes. This result suggests that with the advancement of internet technology and the increased transparency of information, consumers can easily compare service prices and quality across platforms, thus minimizing the impact of financial risk on attitude [83]. Conversely, consistent with Elnadi and Gheith’s study [22] in the car-sharing sector, privacy risk negatively impacts attitudes toward skill-sharing platforms. Users’ concerns about personal information security can lead to negative perceptions of the platform [34]. Therefore, it is crucial for skill-sharing service platforms to establish robust data protection measures. They should implement advanced security technologies to safeguard user data and actively promote these security measures to build trust and confidence among users. As was the case for the findings of Hasan et al. [37] in the car-sharing sector, safety risk also significantly influences attitudes toward skill-sharing service platforms. Consequently, these platforms should implement emergency contact mechanisms, address users’ safety concerns, and ensure that all registered service providers undergo thorough background checks and health screenings. Furthermore, supporting the findings of Ma et al.’s study [84] in the car-sharing sector, functional risk negatively influenced attitudes. Users are concerned about potential discrepancies between advertised and actual services, along with inadequate complaint handling [85]. To address these issues, skill-sharing service platforms must ensure that the quality of service aligns with that advertised and establish effective customer service and complaint resolution processes. By attaching importance to customer feedback, regularly assessing service quality, and continuously optimizing service processes and content, the platforms can guarantee consistency and high standards of user experiences. These measures can effectively manage users’ perceived risks and foster positive attitudes toward platform usage, attracting more users.

Third, corroborating the findings of Zhu et al. [42], this study demonstrated that self-efficacy significantly influences attitudes. Users who feel confident in their ability to navigate these platforms and believe that they can effectively accomplish specific tasks tend to develop more positive consumer attitudes [41]. Thus, skill-sharing service platforms should prioritize boosting user confidence by providing clear and concise usage instructions, facilitating user experiences, and offering appropriate training to enhance users’ self-efficacy. Interestingly, altruistic tendencies did not directly affect attitudes. While this finding diverges from the results of some previous research, it aligns with Hwang and Griffiths’ study [43] on collaborative consumption service. This suggests that although users may self-report high altruistic tendencies and possess personality traits that predispose them to helping others, human behavior is primarily driven by self-interest and practical considerations, such as individual needs and service quality, rather than by motivations to assist service providers [86]. This observation reaffirms that the use of skill-sharing service platforms is predominantly viewed as a transactional activity rather than a charitable or socially supportive endeavor [87].

Fourth, this study identified attitude as the primary factor in predicting continued use intention, corroborating the findings of Zhu et al. [42] in the sharing economy sector. Enhancing user attitudes can effectively promote continued use intention [53]. Consequently, skill-sharing service platforms should implement robust user feedback mechanisms to promptly understand and address user needs and concerns, facilitating continuous improvement. Simultaneously, these platforms should intensify their promotional efforts to ensure that users fully comprehend and leverage the platform’s features and benefits. It is crucial to bolster security measures to safeguard user data, personal safety, and property. By highlighting the platform’s unique advantages and demonstrating tangible user value, skill-sharing services can foster positive user attitudes and strengthen the intention to continue using the platforms.

Finally, this study examined the moderating role of trust and found that it positively influences the relationship between attitude and the intention to continue using the platform. When users have confidence in the platform’s ability to deliver high-quality services, their positive attitudes are more likely to translate into intentions to continue use [67]. Therefore, skill-sharing service platforms should promote their high-quality service processes through traditional mass media and online social media. Encouraging users to share their experiences on social media can effectively build a positive image of the platform in the minds of the users. Such promotional activities increase the platform’s visibility and enhance service reliability, thereby strengthening the impact of attitude on the intention to continue platform use. 

### 5.3. Limitations and Future Research

This study offers valuable insights grounded in the UTAUT model and presents novel perspectives for research on attitude and trust. However, it is important to acknowledge several limitations. Notably, due to the relatively scarce literature on trust within the sharing economy, this study only examined the moderating effect of trust in a single pathway. Future research should expand upon this by considering trust as both a mediating and moderating variable across multiple pathways. This approach would allow for a more thorough exploration of the interrelationships between trust and other variables. Consequently, it could yield deeper insights into trust-related actual behavior, providing more comprehensive theoretical and practical implications for the field.

This study also investigated the intention to continue using the platform. Given the substantial potential user base for skill-sharing services as an emerging technology, the research focused on consumers with platform usage experience, excluding those without such experience. It is important to note that differences in usage behavior may emerge, based on the presence or absence of prior experience, potentially leading to varied research outcomes. Consequently, future studies should consider incorporating individuals without prior usage experience to provide a more comprehensive evaluation of the continued use intention for skill-sharing service platforms. This approach would further enrich and expand the theoretical framework.

Lastly, the sample population of the current study is limited to China. A previous study by Gerlich [88] explored consumer behavior on collaborative economy platforms within EU member states. Additionally, Quattrone et al. [89] conducted a study comparing the adoption of the sharing economy model, specifically focusing on Airbnb, between Western countries and non-Western countries, such as those in Asia and Latin America. In contrast, the sample in the present research is limited to Chinese users, which reduces the scope of the findings. While the results offer valuable insights into the Chinese market, their applicability to other cultures and environments remains unverified. To validate these findings and better understand how these variables operate across diverse cultural contexts, future research should adopt a multicultural approach, incorporating a broader range of participants for a more comprehensive analysis.

## Figures and Tables

**Figure 1 behavsci-14-00765-f001:**
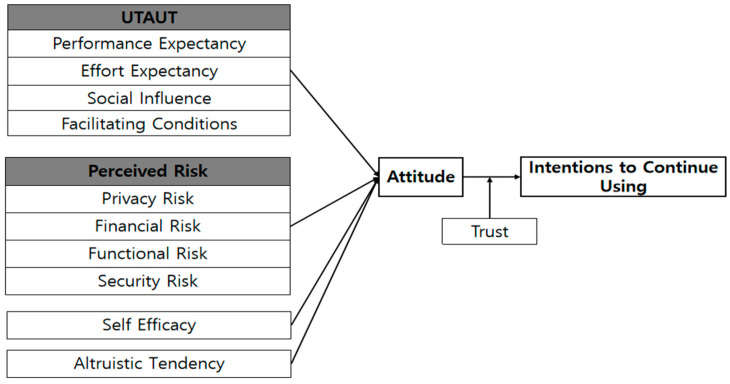
Research model.

**Table 1 behavsci-14-00765-t001:** Survey item composition, according to the UTAUT model.

Measurement Items		
Performance Expectancy (PE)	Skill-sharing service platforms are useful.	[16,22,24]
	Using skill-sharing service platforms helps save time.	
	Using skill-sharing service platforms enhances my work efficiency.	
Effort Expectancy (EE)	Skill-sharing service platforms are easy to use.	
	It is easy to learn how to use skill-sharing service platforms.	
	I am proficient in using skill-sharing service platforms.	
	I quickly adapt to using skill-sharing service platforms.	
Social Influence (SI)	I may try a skill-sharing service platform if people I know recommend it.	
	If many people I know are using a skill-sharing service platform, I may be inclined to try it as well.	
	Seeing information about a skill-sharing service platform in newspapers, on TV, or on social media makes me want to try it.	
Facilitating Conditions (FC)	I meet the requirements for using skill-sharing service platforms.	
	I possess the information or knowledge needed to use skill-sharing service platforms.	
	I can get help through various channels if I face issues with a skill-sharing service platform.	
Privacy Risk (PR)	I am concerned that my personal information might be sold while signing up for a skill-sharing service platform.	[22,34,36]
	I am worried that my personal information could be leaked without my consent while using a skill-sharing service platform.	
	I am concerned that my address might be leaked, increasing the risk of theft, while using a skill-sharing service platform.	
Financial Risk (FIR)	I am concerned that the services provided by skill-sharing service platforms might be more costly than traditional methods.	
	I am concerned that I might not receive discounts when purchasing services through skill-sharing service platforms.	
	I am concerned that the service I receive might not be worth the money I spend on skill-sharing service platforms.	
	I am concerned about potential additional costs when purchasing services through skill-sharing service platforms.	
Functional Risk (FR)	I am concerned that the services advertised by skill-sharing service platforms might differ from the actual services provided.	
	I am concerned that the quality of services I receive from skill-sharing service platforms might not meet my expectations.	
	I am concerned that skill-sharing service platforms might not deliver the level of service I expect.	
	I am concerned that my requests or complaints about the service of skill-sharing service platforms might not be addressed properly.	
Security Risk (SR)	I am concerned that my property might not be safe when using skill-sharing service platforms.	
	I am concerned that services provided by strangers through skill-sharing service platforms might have safety issues.	
	I am concerned about the possibility of criminal activities occurring when using skill-sharing service platforms.	
	I worry about meeting people with infectious diseases through offline services provided by skill-sharing service platforms.	
Self Efficacy (SE)	I am capable of using skill-sharing service platforms well.	[41,42]
	I am confident in utilizing skill-sharing service platforms.	
	I am sure of my ability to accomplish specific tasks well using skill-sharing service platforms.	
Altruistic Tendency (ALT)	Helping others is a rewarding experience.	[46]
	I spend a lot of time helping others.	
	I am eager to help people in difficult situations.	
Trust (TR)	Skill-sharing service platforms are trustworthy.	[54,56]
	Overall, skill-sharing service platforms are reliable.	
	Skill-sharing service platforms provide reliable information.	
Attitude (ATT)	I think using skill-sharing service platforms is wise.	[41,43]
	I think using skill-sharing service platforms is desirable.	
	I think using skill-sharing service platforms is beneficial.	
	I have a positive attitude toward using skill-sharing service platforms for my work.	
	I think using skill-sharing service platforms is a good experience.	
Intentions to Continue Using (ITCU)	I will continue to use skill-sharing service platforms to complete my tasks in the future.	[71]
	I will continue to use skill-sharing service platforms rather than other alternative methods.	
	I will use skill-sharing service platforms frequently in the future.	
	I will increase the time and frequency of using skill-sharing service platforms in the future.	

**Table 2 behavsci-14-00765-t002:** The respondents’ general characteristics.

Classification	Indicators	Frequency	%
Gender	Male	208	50.9
	Female	201	49.1
Age	20–29	102	24.9
	30–39	108	26.4
	40–49	101	24.7
	Over 50 (including 50 years old)	98	24.0
Education level	High school or below	97	23.7
	University or college graduate	209	51.1
	Postgraduate or above	103	25.2
Monthly income	Less than CNY 3000 (excluding CNY 3000)	48	11.7
	CNY 3000 to CNY 6000 (excluding CNY 6000)	130	31.8
	CNY 6000 to CNY 9000 (excluding CNY 9000)	116	28.4
	More than CNY 9000	115	28.1

**Table 3 behavsci-14-00765-t003:** Results of validity and reliability testing of measurement items.

Factor	Variable	Standard Item Loadings	Cronbach’s α	AVE	C.R.
PE1	PE	0.868	0.827	0.626	0.833
PE2		0.731			
PE3		0.768			
EE1	EE	0.794	0.863	0.615	0.864
EE2		0.707			
EE3		0.823			
EE4		0.807			
SI1	SI	0.776	0.813	0.594	0.814
SI2		0.738			
SI3		0.796			
FC1	FC	0.827	0.815	0.602	0.819
FC2		0.773			
FC3		0.723			
PR1	PR	0.803	0.831	0.623	0.832
PR2		0.791			
PR3		0.773			
FIR1	FIR	0.687	0.826	0.547	0.828
FIR2		0.775			
FIR3		0.793			
FIR4		0.698			
FR1	FR	0.808	0.886	0.665	0.888
FR2		0.846			
FR3		0.829			
FR4		0.776			
SR1	SR	0.895	0.922	0.749	0.923
SR2		0.879			
SR3		0.851			
SR4		0.835			
SE1	SE	0.767	0.819	0.608	0.823
SE2		0.819			
SE3		0.751			
ALT1	ALT	0.750	0.797	0.571	0.799
ALT2		0.808			
ALT3		0.705			
TR1	TR	0.887	0.859	0.676	0.862
TR2		0.790			
TR3		0.785			
ATT1	ATT	0.836	0.929	0.724	0.929
ATT2		0.869			
ATT3		0.817			
ATT4		0.868			
ATT5		0.863			
ITCU1	ITCU	0.759	0.860	0.605	0.859
ITCU2		0.794			
ITCU3		0.737			
ITCU4		0.818			
Chi-square = 1166.861, df = 911, χ^2^/df = 1.281; *p* = 0.000, GFI = 0.894; AGFI = 0.875; CFI = 0.975; RMSEA = 0.026					

**Table 4 behavsci-14-00765-t004:** Correlation matrix of measured variables.

	PE	EE	SI	FC	PR	FIR	FR	SR	SE	ALT	TR	ATT	ITCU
PE	0.791												
EE	0.499	0.784											
SI	0.444	0.358	0.771										
FC	0.365	0.377	0.370	0.776									
PR	−0.23	−0.257	−0.385	−0.326	0.789								
FIR	−0.011	−0.018	0.041	−0.048	0.087	0.740							
FR	−0.339	−0.331	−0.339	−0.346	0.408	0.213	0.815						
SR	−0.207	−0.266	−0.323	−0.361	0.380	0.083	0.346	0.865					
SE	0.414	0.287	0.384	0.405	−0.289	−0.028	−0.421	−0.289	0.780				
ALT	0.084	0.006	0.037	0.066	0.024	−0.028	0.046	−0.049	0.090	0.756			
TR	0.397	0.281	0.432	0.355	−0.374	−0.064	−0.437	−0.143	0.455	0.046	0.822		
ATT	0.564	0.530	0.564	0.545	−0.468	−0.007	−0.525	−0.466	0.569	0.006	0.656	0.851	
ITCU	0.326	0.338	0.356	0.345	−0.322	−0.063	−0.352	−0.271	0.385	0.037	0.599	0.635	0.778

Note: diagonal elements are the square root of AVE.

**Table 5 behavsci-14-00765-t005:** Hypothesis testing.

Hypothesis	β	S.E.	C.R.	*p*-Value	Result
H1: PE → ATT	0.173	0.057	3.471	0.000 ***	Accept
H2: EE → ATT	0.160	0.047	3.480	0.000 ***	Accept
H3: SI → ATT	0.161	0.057	3.286	0.001 **	Accept
H4: FC → ATT	0.149	0.052	3.182	0.001 **	Accept
H5: PR → ATT	−0.114	0.045	−2.508	0.012 **	Accept
H6: FIR → ATT	0.050	0.052	1.310	0.190	Reject
H7: FR → ATT	−0.137	0.045	−2.948	0.003 **	Accept
H8: SR → ATT	−0.136	0.039	−3.297	0.000 ***	Accept
H9: SE → ATT	0.204	0.048	4.226	0.000 ***	Accept
H10: ALT → ATT	−0.038	0.061	−0.995	0.320	Reject
H11: ATT → ITCU	0.654	0.054	11.584	0.000 ***	Accept
Chi square = 1032.405, df = 804, χ^2^/df = 1.284; *p* = 0.000, GFI = 0.899; AGFI = 0.881; CFI = 0.976; RMSEA = 0.026					

** *p* < 0.01; *** *p* < 0.001.

**Table 6 behavsci-14-00765-t006:** Verification of the moderating effect of trust.

Hypothesis	Δχ^2^, ∆df	LOW (N = 132)		HIGH (N = 277)		Result
		β	C.R.	β	C.R.	
H12: ATT → ITCU	Δχ^2^ (df = 1) = 7.910 **	0.461 ***	4.361	0.641 ***	8.816	Accept

** *p* < 0.01; *** *p* < 0.001.

## Data Availability

The raw data supporting the conclusions of this article will be made available by the authors on request.

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
