# Peer review of "Chinese Consumers’ Attitudes toward and Intentions to Continue Using Skill-Sharing Service Platforms"

_behavsci, 2024, doi:10.3390/bs14090765_

Round 1
Reviewer 1 Report
Comments and Suggestions for Authors
The skill-sharing service platform is worth a thorough study. The authors expanded UTAUT framework by adding in other variables, accurately explained the transactions on the skill-sharing service platform.
1. Line 189, suggest to illustrate the 4 types of risks and their relevance to the skill-sharing platform
2. Chapter 4 Results could be better explained, especially regarding the moderating role of trust. As trust is defined as confidence towards the platform, it is a common understanding that when one has stronger confidence, one will continue to use the service. Further illustration needed to treat trust as a moderator in this research.
Comments on the Quality of English LanguageEasy to understand
Reviewer 2 Report
Comments and Suggestions for Authors
Dear authors,
Thank you for the opportunity to read your manuscript and to contribute with comments to its improvement. The topic is interesting, although its novelty is rather low. Please see my comments below:
1) I suppose in your study the variable "attitude" in your hypothesis H11 means "positive attitude". Please check again. The hypothesis as stated now would not be correct, as a negative attitude would NOT positively influence the intension. Generally, looking at RQ2, for human behaviour it is normal when a subject "hated" an experience that they will not continue it or vice versa when they "loved" it they will continue using it. It is not clear what the value of RQ2 is and how it would contribute to the field.
2) Used abbreviations must be explained first (see especially line 94-95)
3) Your hypotheses (H5 to H8) represent perceived risks. The hypotheses should be corrected accordingly.
4) Your manuscript often refers to trust. Your H12 includes trust as a variable. Trust has several dimensions which should be discussed in more detail under 2.6. Especially studies on motivators for trust should be discussed.
5) Data collection and sampling: You used voluntary sampling which has advantages abut as well various limitations. You should explain these.
6) As you correctly state, one of the limitations of your study is the geographic limitation. You can improve on that topic by adding a short paragraph on studies from different regions on sharing platforms
Gerlich, M. (2023). The Rise of Collaborative Consumption in EU Member States: Exploring the Impact of Collaborative Economy Platforms on Consumer Behavior and Sustainable Consumption. Sustainability, 15(21), 15491. https://doi.org/10.3390/su152115491
Quattrone, G., Kusek, N. & Capra, L. A global-scale analysis of the sharing economy model – an AirBnB case study. EPJ Data Sci. 11, 36 (2022). https://doi.org/10.1140/epjds/s13688-022-00349-3
Chakravarthi Narasimhan, PurushottamPapatla, Baojun Jiang, PraveenK.,Kopalle, Paul R.,Messinger, SridharMoorthy, et al. Sharing Economy: Review of Current Research and Future Directions. Customer Needs and Solutions.(2018). https://doi.org/10.1007/s40547-017-0079-6
These are just some examples. Many more were conducted.
Overall, a nicely presented manuscript. Especially the data analytics part is well done!
All the best!
Round 2
Reviewer 2 Report
Comments and Suggestions for Authors
Dear author(s)
Thank you for revising the manuscript which seriously improved.
All the best!